# Comparison of the Dynamic Performance of Planar 3-DOF Parallel Manipulators

**Guoning Si** [1] , **Fahui Chen** [1] **and Xuping Zhang** [2],*

[1] School of Health Science and Engineering, University of Shanghai for Science and Technology, Shanghai 200093, China; gnsi@usst.edu.cn (G.S.); 193832311@st.usst.edu.cn (F.C.)
[2] Department of Mechanical and Production Engineering, Aarhus University, 8000 Aarhus, Denmark
* Correspondence: xuzh@mpe.au.dk; Tel.: +45-4189-3167

**Abstract:** This paper presents a comprehensive comparison study on the dynamic performances of three planar 3-DOF parallel manipulators (PPMs): 3-RRR, 3-PRR, and 3-RPR. In this research work, the discrete time transfer matrix method (DT-TMM) is employed for developing dynamic models of the planar parallel manipulators. Numerical simulations using the virtual work principle and ADAMS 2016 software are performed to verify the DT-TMM dynamic model of PPMs. Numerous dynamic performance indices, including dynamic dexterity, the power requirement, energy transmission efficiency, and the joint force/torque margin, are proposed to compare the dynamic performance of three PPMs under the general circular and linear trajectories. The comprehensive analyses and comparisons show that: (1) the 3-RRR PPM has advantages in terms of a circular trajectory, offering the best dynamic dexterity performance, the smallest power requirement, and the second-highest energy transfer efficiency; (2) the 3-PRR PPM performs best in terms of a linear trajectory, offering the best dynamic dexterity, the smallest power requirement range, and the best drive performance; and (3) the 3-RPR PPM has the highest energy transfer efficiency and demonstrates better dynamic performance in a circular trajectory compared to a linear trajectory.

**Keywords:** discrete time transfer matrix method; dynamic modeling; planar parallel manipulator; dynamics dexterity; power requirement; energy transmission efficiency; joint force/torque margin





## 1. Introduction

Parallel manipulators have various advantages over serial manipulators, such as high motion accuracy, large structure stiffness, and the capability of achieving high speeds and undertaking heavy payloads, due to their closed-loop structures [1]. Therefore, 3-DOF planar parallel manipulators (PPMs) have been developed for potential applications in the fields of semiconductor manufacturing, medical surgery, automatic micro-assembly [2,3], etc. Extensive research has been reported in the literature on the kinematics and dynamics of 3-DOF PPMs with identical kinematic chains and symmetrical shapes (3-RRR, 3-PRR, 3-RPR, 3-RPP, 3-RRP, 3-PRP, and 3-PPR) [4–6]. Research efforts were also conducted regarding comparison studies on the kinematic performances of PPMs with the same architecture and with different architectures [7–10]. However, to the best knowledge of the authors, there is no published report of dynamic performance comparison of 3-DOF PPMs. Therefore, this work presents a comprehensive comparison of the dynamic performance of the 3-DOF PPMs, providing valuable guidance and insights into the selection and design of 3-DOF PPMs.

Various dynamic performance indices have been proposed for evaluating the advantages or disadvantages of parallel manipulators in the literature, such as the dynamics manipulability ellipsoid (DME) [11], the generalized inertial ellipsoid (GIE) [12,13], the dynamic condition index (DCI) [14,15], the global dynamic condition index (GDCI) [16–18], motion/force transmissibility [19], energy transfer efficiency, the driving force/torque balance [20], the joint force/torque margin [21,22], and power and energy consumption [22,23].

In terms of the dynamic modeling of PPMs in the current literature, the dynamics equations of PPMs were usually established using traditional dynamic modeling methods, such as the Lagrange equation [24], the Newton–Euler equation [25], the virtual work principle [26], and the Kane method [27]. On the other hand, the discrete time-transfer matrix method (DT-TMM) is a recently developed method for modeling the dynamics of multi-body systems, and its high computational efficiency provides a powerful tool for the study of multi-body system dynamics. Compared with the virtual work principle, DT-TMM does not need to establish global dynamic equations for each PPM; it only needs to build a library of transfer matrices for the components to assemble system transfer matrices for different configurations, so it is highly programmatic and offers modeling flexibility [28–30]. Moreover, the matrices involved in DT-TMM are always small, and the orders of the matrices do not increase with the degrees of freedom of the multi-body system, so the computational cost of dynamic analysis can be significantly reduced [29–31].

Therefore, the dynamics models of 3-RRR, 3-PRR, and 3-RPR PPMs are established based on DT-TMM in this paper. Then, the system is verified with numerical simulations, using the models from the virtual work principle and ADAMS. In engineering applications, the dynamic dexterity of the PPM is most commonly used to evaluate the acceleration/deceleration capability of the PPM in all directions; power and energy are directly related to the cost and efficiency of the PPM, and the driving performance of the actuated joints is critical to the application of the PPM. Therefore, the dexterity dynamics, the power requirement, the energy transmission efficiency, and the joint force/torque margin are proposed as dynamic performance indices to compare the dynamic performance among the three PPMs under different trajectory simulation motions. This comparative study of the dynamic performance of the three different architectures of 3-DOF PPMs can provide a theoretical basis for the selection and design of 3-DOF PPMs.

## 2. Coordinate System and Manipulator Architecture Description of Three PPMs

The architectures of the 3-RRR, 3-PRR, and 3-RPR PPMs illustrated in Figure 1 are composed of a fixed base and a mobile platform that is linked to the base by three independent planar kinematical chains. The inertial coordinate system $O - XY$ is located at the center of the base platform. The fixed base platform $A_1A_2A_3$ and mobile platform $C_1C_2C_3$ of each PPM have the shape of an equilateral triangle, and the width of the base platform and the mobile platform are denoted by $L_B$ and $L_P$, respectively. The position of the platform at its mass center $G$ is defined as $(x_G, y_G)$, and the orientation of the mobile platform is denoted by $\varnothing$. Each chain of three PPMs is composed of three joints mounted in sequence, including one active joint in each chain. Each kinematic chain of the 3-RRR PPM in Figure 1a comprises three revolute joints and two links, and the first revolute joint is actuated. As shown in Figure 1b, each chain of the 3-PRR PPM consists of a prismatic joint, two revolute joints, a slider, and a link, and the prismatic joint is actuated. Similarly, each chain of the 3-RPR PPM illustrated in Figure 1c has two rotating joints, a prismatic joint, and a telescopic link, and the first revolute joint is actuated. In this work, the length of links $A_iB_i$ and $B_iC_i$ in the 3-RRR PPM is denoted by $l_{i1}$ and $l_{i2}$, respectively; the length of the slider rail $A_iB_i$ and the length of links $B_iC_i$ in 3-PRR PPMs are denoted by $\rho_i$ and $l_{i3}$, respectively; the length of three variable-length links $A_iC_i$ with the same topology in 3-RPR PPM is denoted by $l_{i4}$, where $i = 1, 2, 3$.

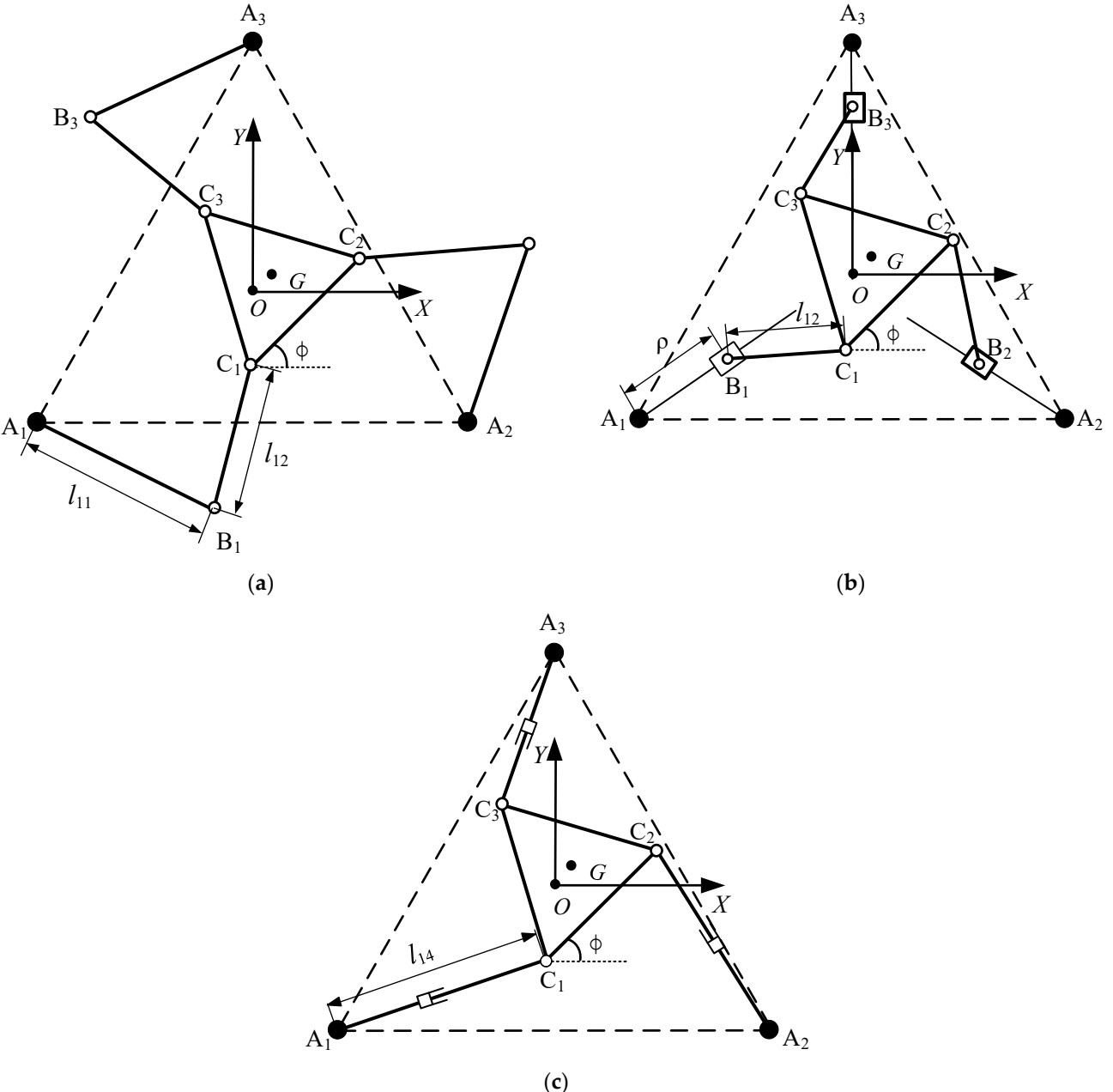

**Figure 1.** Coordinate system and manipulator architectures of three PPMs: (**a**) 3-RRR PPM, (**b**) 3-PRR PPM, (**c**) 3-RPR PPM.

## 3. Dynamic Model and Simulations of Three PPMs

This section establishes the dynamic equations of the 3-RRR, 3-PRR, and 3-RPR PPMs, based on the discrete time transfer matrix method (DT-TMM). To verify the effectiveness of the DT-TMM, the dynamic models are also developed based on the virtual work principle and ADAMS 2016 software. The forces and torques calculated from the three methods are compared with the numerical simulations.

### 3.1. Discrete Time Transfer Matrix Method

The general theorems and steps of DT-TMM are similar to those reported in [30]. The dynamic modeling of the three PPMs, based on DT-TMM, includes six steps:

(1) Performing system discretization, each PPM is divided into subsystems, which can be represented by individual components, such as links, sliders and hinges;

(2) Defining a state vector at both the inboard end and outboard end for each component;

(3) Establishing kinematic and dynamic equations of components and linearizing the kinematic and dynamic equations of the components;

(4) Deriving the component transfer matrix of each component based on its linearized kinematic and dynamic equations;

(5) Assembling the component transfer matrix to obtain the transfer equation for each subsystem and then the global transfer equation with the global transfer matrix of the whole system;

(6) Applying the boundary conditions to the state vectors of the system and calculating the unknown quantities.

### 3.1.1. State Vector and Transfer Equation

The state vectors in the transfer matrix are essential information about objects at the connection points. The state vector of the connection point between two components is defined as:

$$z = \begin{bmatrix} x, & y, & \theta, & M_z, & q_x, & q_y, & 1 \end{bmatrix}^T \tag{1}$$

where $x$, $y$, and $\theta$ are the displacement and orientation of the connection point for the global inertial coordinate system and the orientation angle of the body; $M_z$, $q_x$, $q_y$ are the corresponding internal torques and internal forces in the same reference system, respectively.

By linearizing the kinematic and dynamic equations of the component, the transfer matrix of a single component can be calculated as:

$$z_{i,i+1}(t_i) = U_i(t_i)z_{i,i-1}(t_i) \tag{2}$$

where the transfer matrix of the $i$th component $U_i$, $(i = 1, 2 \ldots n)$ is the functions of the motion quantities known at time instant $t_i$, and the transfer matrix expresses the transformation between the state vectors at its outboard end $z_{i,i+1}$ and inboard end $z_{i,i-1}$.

Connecting all the components of the system, the overall system transfer equation can be assembled and calculated as:

$$z_n = U_{sys}z_1 \tag{3}$$

where $U_{sys} = U_n U_{n-1} \ldots U_2 U_1$ is the overall transfer matrix, and $U_1, U_2, \cdots, U_n$ are all the transfer matrixes of the components in the system.

### 3.1.2. Transfer Matrices of Components

Linearization is an important step in establishing a transfer matrix, via linearization methods such as the Newmark-β numerical integration method; the Taylor expansion of trigonometric function and the polynomial expansion method are described in detail in the study by [29]. The detailed derivation of transfer matrices for rigid bodies and joints in planar motion has been described in [28–34]. The transfer matrices of a link and smooth hinge were given by the authors of [30,31], and the transfer matrices of the slider can be found in [35]. This section will focus on the transfer matrix of the mobile platform.

In this work, the mobile platform is a rigid body with three inboard ends and one outboard end. In the case of a multi-inboard and multi-outboard rigid body, only the matrix relationship between the inboard and outboard ends can be obtained; the transfer equation from one end to the other end cannot be written directly.

As shown in Figure 2, the mobile platform has three inboard ends $I_1$, $I_2$, and $I_3$, and an outboard end $O_G$ at the mass center of the mobile platform. The inertial coordinate system is $O - XY$. The moving coordinate system $O_1 - x_1 y_1$ is fixed at the first inboard end and parallel to the inertial coordinate system. The coordinate system $O_2 - x_2 y_2$ is the moving coordinate system of the mobile platform, with a rotation angle of $\theta$ at the inboard end $I_1$. The position coordinates for inboards and outboards with respect to the coordinate are $(a_{11}, a_{21})$, $(a_{12}, a_{22})$, $(a_{13}, a_{23})$, and $(b_1, b_2)$. Considering that this mobile platform is a rigid body, the angular relationship can be calculated as:

$$\theta_{I_j} = \theta_{I_1} (j = 2, 3) \tag{4}$$

$$\theta_O = \theta_{I_1} \tag{5}$$

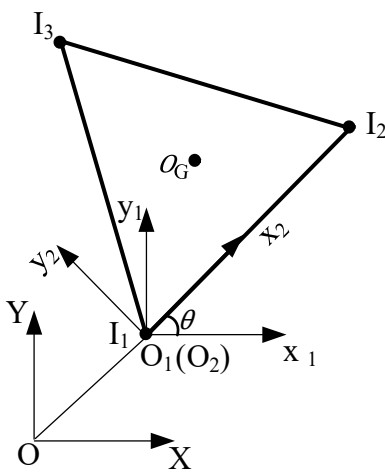

**Figure 2.** Mobile platform of PPM.

The positions of the inboard and outboard ends, in terms of $I_1$ and $\theta_{I_1}$ at the time step $t = i$ can be calculated as:

$$\begin{bmatrix} x_{I_j} \\ y_{I_j} \end{bmatrix} = \begin{bmatrix} x_{I_1} \\ y_{I_1} \end{bmatrix} + \begin{bmatrix} -y_{I_1 I_j}(t_{i-1}) \\ x_{I_1 I_j}(t_{i-1}) \end{bmatrix} \theta_{I_1}(t_i) + \begin{bmatrix} a_{1,j}\overline{S} - a_{2,j}\overline{S} \\ a_{1,j}\overline{C} - a_{2,j}\overline{S} \end{bmatrix} \tag{6}$$

$$\begin{bmatrix} x_{O_G} \\ y_{O_G} \end{bmatrix} = \begin{bmatrix} x_{I_1} \\ y_{I_1} \end{bmatrix} + \begin{bmatrix} -y_{I_1 O_G}(t_{i-1}) \\ x_{I_1 O_G}(t_{i-1}) \end{bmatrix} \theta_{I_1}(t_i) + \begin{bmatrix} b_1\overline{S} - b_2\overline{C} \\ b_1\overline{C} - b_2\overline{S} \end{bmatrix} \tag{7}$$

where:

$$x_{I_1 I_j} = a_{1,j}c_{I_1} - a_{2,j}s_{I_1} \approx a_{1,j}\overline{c} - a_{2,j}\overline{s} \tag{8}$$

$$y_{I_1 I_j} = a_{1,j}s_{I_1} + a_{2,j}c_{I_1} \approx a_{1,j}\overline{s} + a_{2,j}\overline{c} \tag{9}$$

$$x_{I_1 O_G} = b_1 c_{I_1} - b_2 s_{I_1} \approx b_1\overline{c} - b_2\overline{s} \tag{10}$$

$$y_{I_1 O_G} = b_1 s_{I_1} + b_2 c_{I_1} \approx b_1\overline{s} + b_2\overline{c} \tag{11}$$

where $s_I = \sin\theta_I$, $c_I = \cos\theta_I$, $\overline{c}$, $\overline{s}$ and $\overline{C}$, $\overline{S}$ are the linearization of trigonometric terms that can be obtained from the previous study by [28]. Here, $(x_{I_j}, y_{I_j})$ and $(x_{O_G}, y_{O_G})$ are the position coordinates of inboards $I_j$ and outboards $O_G$ in the inertial coordinate system.

The dynamic equations for the mobile platform in the $x$ and $y$ directions are calculated as:

$$\sum_{j=1}^{3} q_{x_{I_j}} - q_{x_{O_G}} + f_{x_G} = m_G \ddot{x}_G \tag{12}$$

$$\sum_{j=1}^{3} q_{y_{I_j}} - q_{y_{O_G}} + f_{y_G} = m_G \ddot{y}_G \tag{13}$$

where $q_{x_{I_j}}$, $q_{y_{I_j}}$ are internal forces acting at the inboard ends, while $q_{x_{O_G}}$, $q_{y_{O_G}}$ are internal forces acting at the outboard end. $f_{x_G}$ and $f_{y_G}$ are external forces applied to the mass center. $m_G$ is the mass of the mobile platform, $\ddot{x}_G$ and $\ddot{y}_G$ are accelerations on the mass center of the mobile platform.

Considering the moment balance, the rotational equations about the inboard $I_1$ is expressed as:

$$\sum M = \frac{dG_{I_1}}{dt} + m_G r_{I_1 G} \times a_{I_1}. \tag{14}$$

Projecting Equation (14) onto the inertial coordinate system, it can be calculated as:

$$
\begin{aligned}
J_{I_1}\ddot{\theta}_I + m_G x_{I_1 G}\ddot{y}_{I_1} \quad - m_G y_{I_1 G}\ddot{x}_{I_1} \\
= -\sum_{j=1}^{3} m_{I_j} + m_O + m_G - \sum_{j=1}^{3} q_{x_{I_j}} y_{I_1 I_j} + \sum_{j=1}^{3} q_{y_{I_j}} x_{I_1 I_j} \\
+ q_{x_{O_G}} y_{I_1 O_G} + q_{y_{O_G}} x_{I_1 O_G} - f_{x_G} y_{I_1 O_G} + f_{y_G} x_{I_1 O_G}
\end{aligned}
\tag{15}
$$

where $G_{I_1} = J_{I_1}\dot{\theta}_I$ is the absolute moment of momentum of the mobile platform with respect to $I_1$, and $J_{I_1}$ is the rotational inertia of the mobile platform with respect to $I_1$. $r_{I_1 G}$ is the vector diameter of the mobile platform center $G$, relative to point $I_1$, and $a_{I_1}$ is the absolute acceleration of point $I_1$.

Writing Equations (4) and (6) in matrix form gives the result:

$$
U_{I_j I_1} z_{I_1} + U_{I_1} z_{I_j} = 0_{3\times 1} \quad (j = 2, 3)
\tag{16}
$$

where:

$$
U_{I_j I_1} =
\begin{bmatrix}
1 & 0 & -y_{I_1 I_j}(t_{i-1}) & 0 & 0 & 0 & a_{1,j}\overline{S} - a_{2,j}\overline{S} \\
0 & 1 & x_{I_1 I_j}(t_{i-1}) & 0 & 0 & 0 & a_{1,j}\overline{C} - a_{2,j}\overline{S} \\
0 & 0 & 1 & 0 & 0 & 0 & 0
\end{bmatrix}
\tag{17}
$$

$$
U_{I_1} = \begin{bmatrix} -I_3 & O_{3\times 4} \end{bmatrix}
\tag{18}
$$

Similarly, Equations (5) and (7) are written in matrix form as:

$$
U_{O_G I_1} z_{I_1} + U_{I_1} z_{O_G} = 0_{3\times 1}
\tag{19}
$$

Linearizing Equations (12), (13), and (15), we write them in matrix form as:

$$
U_{I_1}^4 z_{I_1} + U_{I_2}^4 z_{I_2} + U_{I_3}^4 z_{I_3} + U_{O_G}^4 z_{O_G} = 0_{3\times 1}
\tag{20}
$$

Thus, we define the state vector for the mobile platform with three inboard ends and one outboard end as:

$$
z_{tot} = \begin{bmatrix} z_{I_1}^T & z_{I_2}^T & z_{I_3}^T & z_{O_G}^T \end{bmatrix}^T
\tag{21}
$$

where $z_{I_1}$, $z_{I_2}$ and $z_{I_3}$ are the three state vectors of the inboard ends, and $z_{O_G}$ is the outboard state vector. From the Equations (16), (19) and (20), the transfer equation of the mobile platform can be calculated as:

$$
U_P z_{tot} = 0_{12\times 1}
\tag{22}
$$

where:

$$
U_P =
\begin{bmatrix}
U_{I_2 I_1} & U_{I_1} & O_{3\times 7} & O_{3\times 7} \\
U_{I_3 I_1} & O_{3\times 7} & U_{I_1} & O_{3\times 7} \\
U_{O_G I_1} & O_{3\times 7} & O_{3\times 7} & U_{I_1} \\
U_{I_1}^4 & U_{I_2}^4 & U_{I_3}^4 & U_{O_G}^4
\end{bmatrix}
\tag{23}
$$

where the $U_{I_j I_1}$, $U_{I_1}$, $U_{O_1 I_1}$, $U_{I_j}^4 (j = 2, 3)$ and $U_{I_1}^4$ in the transfer matrix $U_P$ and the detailed derivation of kinematic and dynamic equations are given in [34].

### 3.2. Dynamic Model of Three PPMs with DT-TMM

The DT-TMM models of 3-RRR, 3-PRR, and 3-RPR PPMs illustrated in Figures 3–5 are all chain systems with planar motions, and each PPM can be divided into three branches: I, II, and III. Each branch can be considered as a chain subsystem consisting of rigid bodies (links and sliders) and smooth hinges. These chain subsystems are connected to the three inboard ends on the moving platform by smooth hinges. For calculation purposes, each component of the system is represented by a corresponding number in Figures 3–5.

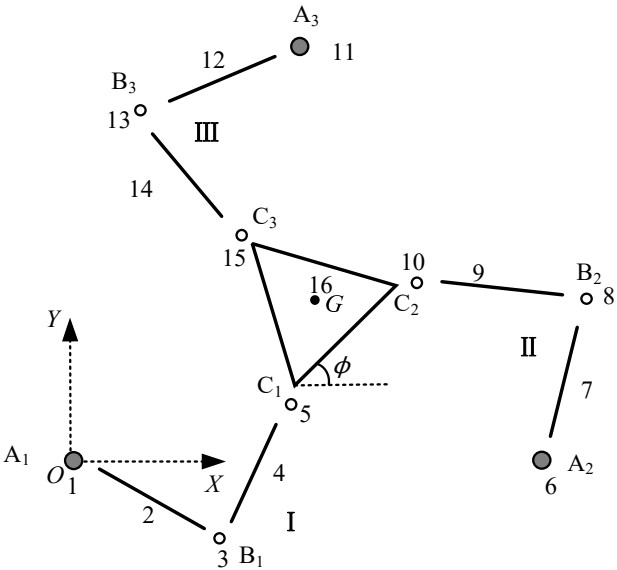

**Figure 3.** DT-TMM model of 3-RRR PPM.

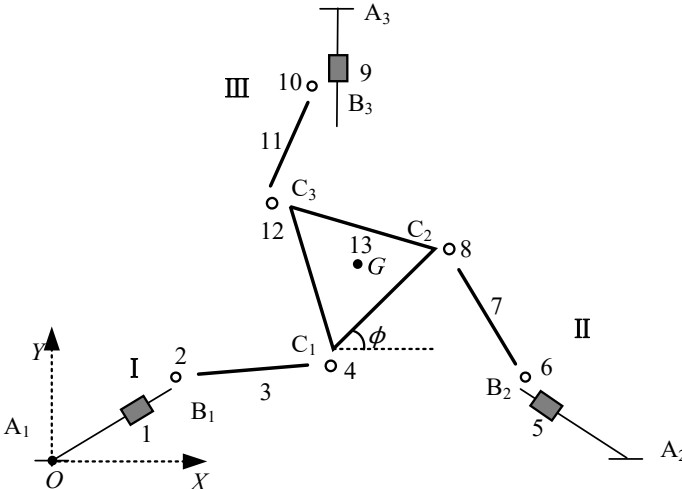

**Figure 4.** DT-TMM model of 3-PRR PPM.

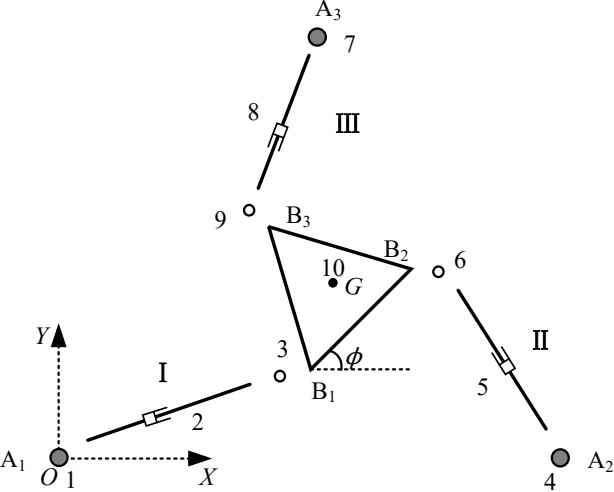

**Figure 5.** DT-TMM model of 3-RPR PPM.

3.2.1. DT-TMM Model of 3-RRR PPM

In this case, the 3-RRR PPM has been divided into three chain subsystems, I, II and III, as illustrated in Figure 3, each chain represented by three smooth hinges and two links. In total, the 3-RRR PPM consists of 16 components, as marked in Figure 3.

The three state vectors of the inboard ends of the three-chain subsystem in 3-RRR PPM are written as:

$$
\begin{cases}
z_{I_1 rrr} = \begin{bmatrix} x \ y \ \theta \ M \ q_x \ q_y \ 1 \end{bmatrix}^T_{I_1} \\
z_{I_6 rrr} = \begin{bmatrix} x \ y \ \theta \ M \ q_x \ q_y \ 1 \end{bmatrix}^T_{I_2} \\
z_{I_{11} rrr} = \begin{bmatrix} x \ y \ \theta \ M \ q_x \ q_y \ 1 \end{bmatrix}^T_{I_{11}}
\end{cases}
\tag{24}
$$

The outboard end of the system is at the mass center $G$ of the mobile platform; it has a state vector in the form of:

$$
z_{Grrr} = \begin{bmatrix} x \ y \ \theta \ M \ q_x \ q_y \ 1 \end{bmatrix}^T_G
\tag{25}
$$

The mobile platform has three state vectors of the inboard ends, which are also the outboard state vectors of the three chain subsystems, defined as $z_{O_5 rrr}$, $z_{O_{10} rrr}$ and $z_{O_{15} rrr}$, and one state vector of the outboard end is $z_{Grrr}$. The transfer equation is expressed as:

$$
U_{16} \begin{bmatrix} z^T_{O_5 rrr} & z^T_{O_{10} rrr} & z^T_{O_{15} rrr} & z^T_{Grrr} \end{bmatrix}^T = 0_{12 \times 1}
\tag{26}
$$

where $U_{16}$ is the transfer matrix of the mobile platform, which can be found via Equation (23). The transfer equations for each of the three chain subsystems in the 3-RRR PPM can be expressed as:

$$
\begin{cases}
z_{O_5 rrr} = U_{Irrr} z_{I_1 rrr} \\
z_{O_{10} rrr} = U_{IIrrr} z_{6rrr} \\
z_{O_{15} rrr} = U_{IIIrrr} z_{I_{11} rrr}
\end{cases}
\tag{27}
$$

where $U_{Irrr} = U_5 U_4 U_3 U_2 U_1$, $U_{IIrrr} = U_{10} U_9 U_8 U_7 U_6$, $U_{IIIrrr} = U_{15} U_{14} U_{13} U_{12} U_{11}$. The value $U_i$ $(i = 2, 3, \ldots, 15)$ is the transfer matrix of component $i$ in of 3-RRR PPM, substituting Equation (27) into Equation (26). Hence, Equation (26) can be rewritten as:

$$
U_{16rrr} \begin{bmatrix} z_{O_5} \\ z_{O_{10}} \\ z_{O_{15}} \\ z_G \end{bmatrix}_{rrr} = U_{16} \begin{bmatrix} U_{Irrr} & 0_{7\times 7} & 0_{7\times 7} & 0_{7\times 7} \\ 0_{7\times 7} & U_{IIrrr} & 0_{7\times 7} & 0_{7\times 7} \\ 0_{7\times 7} & 0_{7\times 7} & U_{IIIrrr} & 0_{7\times 7} \\ 0_{7\times 7} & 0_{7\times 7} & 0_{7\times 7} & I_{7\times 7} \end{bmatrix} \begin{bmatrix} z_{I_1} \\ z_{I_6} \\ z_{I_{11}} \\ z_G \end{bmatrix}_{rrr} = 0_{12\times 1}
\tag{28}
$$

The boundary conditions can be obtained from:

$$
\begin{cases}
z_{I_1 rrr}(1) = x_{A1rrr} & z_{I_1 rrr}(2) = y_{A1rrr} \\
z_{I_6 rrr}(1) = x_{A2rrr} & z_{I_6 rrr}(2) = y_{A2rrr} \\
z_{I_{11} rrr}(1) = x_{A3rrr} & z_{I_{11} rrr}(2) = y_{A3rrr} \\
z_{Grrr}(1) = x_G & z_{Grrr}(2) = y_G & z_{Grrr}(3) = \varnothing \\
z_{Grrr}(4) = 0 & z_{Grrr}(5) = 0 & z_{Grrr}(6) = 0
\end{cases}
\tag{29}
$$

where $z(i)$, $i = 1, 2, ..6$, represents the $i$th state variable in the state vector $z$, $x_{Airrr}$ and $y_{Airrr}$ $(i = 1, 2, 3)$ represent the position of $A_{irrr}$, $x_G$ and $y_G$ are the positions of the platform at its mass center $G$. Substituting the boundary conditions into Equations (24) and (25), the driving torque in the inboard state can be established with Equation (28).

3.2.2. DT-TMM Model of 3-PRR PPM

As shown in Figure 4, each chain subsystem of the 3-PRR PPM consists of a slider, a link, and two smooth hinges. In total, the 3-RRR PPM has 13 components, as marked in Figure 4.

Similarly, since the sliders are moving in a constant direction and there is no external torque acting on them, the inboard state vectors for each chain subsystem of 3-PRR PPM can be expressed as:

$$
\begin{cases}
z_{I_1\,prr} = \begin{bmatrix} x\ y\ 0\ 0\ q_x\ q_y\ 1 \end{bmatrix}^T{}_{I_1\,prr} \\
z_{I_5\,prr} = \begin{bmatrix} x\ y\ 0\ 0\ q_x\ q_y\ 1 \end{bmatrix}^T{}_{I_5\,prr} \\
z_{I_9\,prr} = \begin{bmatrix} x\ y\ 0\ 0\ q_x\ q_y\ 1 \end{bmatrix}^T{}_{I_9\,prr}
\end{cases}
\tag{30}
$$

The transfer equations for each chain system of 3-PRR PPMs can be obtained from:

$$
\begin{cases}
z_{O_4\,prr} = U_{I\,prr} z_{I_1\,prr} \\
z_{O_8\,prr} = U_{II\,prr} z_{I_5\,prr} \\
z_{O_{12}\,prr} = U_{III\,prr} z_{I_9\,prr}
\end{cases}
\tag{31}
$$

where $U_{I\,prr} = U_4 U_3 U_2 U_1$, $U_{II\,prr} = U_8 U_7 U_6 U_5$, and $U_{III\,prr} = U_{12} U_{11} U_{10} U_9$. $U_i\ (i = 2, 3, \dots, 13)$ is the transfer matrix of component $i$ in the 3-PRR PPM. A detailed derivation of the component transfer matrix involved in 3-PRR PPM can be obtained from the study by [35].

The dynamic equations of the platform, based on the Newton–Euler equation, can be written as:

$$
\sum F_{x\,prr} = \ddot{x}_p m_p, \ \sum F_{y\,prr} = \ddot{y}_p m_p, \sum M_{prr} = \ddot{\varnothing}_p I_p
\tag{32}
$$

Using kinematic equations, Equation (32) can be combined and rewritten in matrix form as:

$$
\begin{bmatrix}
1 & 0 & 1 & 0 & 1 & 0 \\
0 & 1 & 0 & 1 & 0 & 1 \\
-d_{y1} & d_{x1} & -d_{y2} & d_{x2} & -d_{y3} & d_{x3}
\end{bmatrix}
\begin{bmatrix}
q_{x4} \\ q_{y4} \\ q_{x8} \\ q_{y8} \\ q_{x12} \\ q_{y12}
\end{bmatrix}_{prr}
=
\begin{bmatrix}
\ddot{x}_G m_p \\ \ddot{y}_G m_p \\ \ddot{\varnothing} I_p
\end{bmatrix}_{prr}
\tag{33}
$$

where $d_{xi} = -\frac{\sqrt{3}}{3} L_P \cos \gamma_i (i = 1, 2, 3)$, $d_{yi} = \frac{\sqrt{3}}{3} L_P \sin \gamma_i (i = 1, 2, 3)$, $\begin{bmatrix} \gamma_1 & \gamma_2 & \gamma_3 \end{bmatrix} = \begin{bmatrix} 210 & -30 & 90 \end{bmatrix}$; $q_{xi}$, $q_{yi}(i = 4, 8, 12)$ are internal forces at the outboard tips of linkages.

To solve the six unknowns given by the dynamic Equation (33), it is necessary to establish the relationships of the internal forces, $q_{xi}$ and $q_{yi}$. The transfer Equation (31) for each chain system can be rewritten as:

$$
\begin{bmatrix}
x \\ y \\ 0 \\ 0 \\ q_x \\ q_y \\ 1
\end{bmatrix}_{4,8,12\,prr}
=
\begin{bmatrix}
u_{11} & u_{12} & u_{13} & u_{14} & u_{15} & u_{16} & u_{17} \\
u_{21} & u_{22} & u_{23} & u_{24} & u_{25} & u_{26} & u_{27} \\
u_{31} & u_{32} & u_{33} & u_{34} & u_{35} & u_{36} & u_{37} \\
u_{41} & u_{42} & u_{43} & u_{44} & u_{45} & u_{46} & u_{47} \\
u_{51} & u_{52} & u_{53} & u_{54} & u_{55} & u_{56} & u_{57} \\
u_{51} & u_{52} & u_{53} & u_{54} & u_{55} & u_{56} & u_{57} \\
u_{61} & u_{62} & u_{63} & u_{64} & u_{65} & u_{66} & u_{67} \\
0 & 0 & 0 & 0 & 0 & 0 & 1
\end{bmatrix}_{I,II,III\,prr}
\begin{bmatrix}
x \\ y \\ 0 \\ 0 \\ q_x \\ q_y \\ 1
\end{bmatrix}_{1,5,9\,prr}
\tag{34}
$$

The fourth row of the transfer matrix provides the moment balance relationship, so that the internal forces along the $y$-direction at the inboard end of 1, 5, and 9 can be written as the functions of the internal forces along the $x$-direction, shown as:

$$
q_{yi\,prr} = -\frac{1}{u_{46,i}} \begin{bmatrix} u_{41} & u_{42} & u_{47} \end{bmatrix}_i \begin{bmatrix} x \\ y \\ 1 \end{bmatrix}_i - \frac{u_{45,i}}{u_{46,i}} q_{xi\,prr}, i = 1,\ 5,\ 9
\tag{35}
$$

Inserting Equation (35) into the fifth and sixth rows in Equation (34), $q_x$, and $q_y$ at point $i + 3$ (where $i = 1, 5, 9$) can be expressed only as the functions of the inboard internal forces $q_{xi}$:

$$
\begin{bmatrix} q_x \\ q_y \end{bmatrix}_{i+3\,prr} = \begin{bmatrix} u_{51} & u_{52} & u_{53} & u_{54} & u_{55} & u_{56} & u_{57} \\ u_{61} & u_{62} & u_{63} & u_{64} & u_{65} & u_{66} & u_{67} \end{bmatrix}_{i\,prr} \begin{bmatrix} q_x \\ q_y(q_x) \end{bmatrix}_{i\,prr} \tag{36}
$$

where $q_y(q_x)$ represents the fact that $q_y$ is a function of $q_x$ at the inboard end $i$. After substituting Equation (36) into Equation (33), it can be rewritten as:

$$
\begin{bmatrix} 1 & 0 & 1 & 0 & 1 & 0 \\ 0 & 1 & 0 & 1 & 0 & 1 \\ -d_{y1} & d_{x1} & -d_{y2} & d_{x2} & -d_{y3} & d_{x3} \end{bmatrix}_{prr} \begin{bmatrix} q_{y4}(q_{x1}) \\ q_{y4}(q_{x1}) \\ q_{x8}(q_{x5}) \\ q_{y8}(q_{x5}) \\ q_{x12}(q_{x9}) \\ q_{y12}(q_{x9}) \end{bmatrix}_{prr} = \begin{bmatrix} \ddot{x}_G m_p \\ \ddot{y}_G m_p \\ \ddot{\varnothing} I_p \end{bmatrix}_{prr} \tag{37}
$$

Equation (37) contains only three unknowns; thus, the actuated forces on the *x*-direction and *y*-direction that are applied to the sliders can be obtained from Equations (35) and (37).

### 3.2.3. DT-TMM Model of 3-RPR PPM

Using Figure 1c, the closed-loop geometric equation of each chain can be expressed as:

$$
\overrightarrow{OA_{irpr}} + \overrightarrow{A_iC_{irpr}} = \overrightarrow{OG_{rpr}} + \overrightarrow{GC_{irpr}} \tag{38}
$$

The vector in Equation (38) is projected onto the X and Y axis, and can be rewritten as:

$$
\begin{cases} l_{4i} \cos \theta_{irpr} = x_G - x_{Airpr} + r \cos (\gamma_i + \varnothing) \\ l_{4i} \sin \theta_{irpr} = y_G - y_{Airpr} + r \sin (\gamma_i + \varnothing) \end{cases} \tag{39}
$$

where $(x_{Airpr}, y_{Airpr})$ are the positions of point $A_{irpr}$, and $\theta_{irpr}$ is the rotation angle of $A_iC_{irpr}$. $r$ is the length of $G_iC_{irpr}$, and $\gamma_i = \frac{4\pi i - 3\pi}{6}$. The organizing equation to eliminate $\theta_{irpr}$, $l_{4i}$ can be written as:

$$
l_{4i}^2 = (x_G - x_{Airpr} + r \cos (\gamma_i + \varnothing))^2 + (y_G - y_{Airpr} + r \sin (\gamma_i + \varnothing))^2 \tag{40}
$$

As shown in Figure 5, the dynamic model of 3-RPR PPM has been divided into three chain subsystems, with each chain represented by two smooth hinges and a variable length link. In total, the 3-RRR PPM consists of 10 components, as marked in Figure 5.

The three state vectors of the inboard ends of three chain subsystems in the 3-RPR PPM are written as:

$$
\begin{cases} z_{I_1 rpr} = \begin{bmatrix} x & y & \theta & M & q_x & q_y & 1 \end{bmatrix}^T_{I_1 rpr} \\ z_{I_4 rpr} = \begin{bmatrix} x & y & \theta & M & q_x & q_y & 1 \end{bmatrix}^T_{I_4 rpr} \\ z_{I_7 rpr} = \begin{bmatrix} x & y & \theta & M & q_x & q_y & 1 \end{bmatrix}^T_{I_7 rpr} \end{cases} \tag{41}
$$

The outboard end of the system is at the mass center $G$ of the moving platform; it has a state vector in the form of:

$$
z_{Grpr} = \begin{bmatrix} x & y & \theta & M & q_x & q_y & 1 \end{bmatrix}^T_{Grpr} \tag{42}
$$

Similar to 3-RRR PPM, three state vectors of the mobile platform inboard ends are defined as $z_{O_3 rpr}$, $z_{O_6 rpr}$ and $z_{O_9 rpr}$, and one state vector of the outboard end is $z_G$; the transfer equation is written as:

$$U_{10rpr} \begin{bmatrix} z_{O_3}^T & _{rpr} & z_{O_6}^T & _{rpr} & z_{O_9}^T & _{rpr} & z_{Grpr}^T \end{bmatrix}^T = \mathbf{0}_{12 \times 1} \tag{43}$$

where $U_{10rpr}$ is the transfer matrix of the mobile platform. The transfer equations for three chain subsystems of the 3-RPR PPM are written as:

$$\begin{cases} z_{O_3 rpr} = U_3 U_2 U_1 z_{I_1 rpr} = U_{\text{I}rpr} z_{I_1 rpr} \\ z_{O_6 rpr} = U_6 U_5 U_4 z_{I_4 rpr} = U_{\text{II}rpr} z_{I_4 rpr} \\ z_{O_9 rpr} = U_9 U_8 U_7 z_{I_7 rpr} = U_{\text{III}rpr} z_{I_7 rpr} \end{cases} \tag{44}$$

By substituting Equation (44) into Equation (43), hence, Equation (43) can be rewritten as:

$$U_{10rpr} \begin{bmatrix} z_{O_3} \\ z_{O_6} \\ z_{O_9} \\ z_G \end{bmatrix}_{rpr} U_{10rpr} \begin{bmatrix} U_{\text{I}rpr} & \mathbf{0}_{7\times7} & \mathbf{0}_{7\times7} & \mathbf{0}_{7\times7} \\ \mathbf{0}_{7\times7} & U_{\text{II}rpr} & \mathbf{0}_{7\times7} & \mathbf{0}_{7\times7} \\ \mathbf{0}_{7\times7} & \mathbf{0}_{7\times7} & U_{\text{III}rpr} & \mathbf{0}_{7\times7} \\ \mathbf{0}_{7\times7} & \mathbf{0}_{7\times7} & \mathbf{0}_{7\times7} & I_{7\times7} \end{bmatrix} \begin{bmatrix} z_{I_1} \\ z_{I_4} \\ z_{I_7} \\ z_G \end{bmatrix}_{rpr} = \mathbf{0}_{12\times1} \tag{45}$$

The initial condition and boundary conditions can be written as:

$$\begin{cases} z_{I_1 rpr}(1) = x_{A1rpr} & z_{I_1 rpr}(2) = y_{A1rpr} \\ z_{I_4 rpr}(1) = x_{A2rpr} & z_{I_4 rpr}(2) = y_{A2rpr} \\ z_{I_7 rpr}(1) = x_{A3rpr} & z_{I_7 rpr}(2) = y_{A3rpr} \\ z_{Grpr}(1) = x_{Grpr} & z_{Grpr}(2) = y_{G \ rpr} & z_{Grpr}(3) = \varnothing \\ z_{Grpr}(4) = 0 & z_{Grpr}(5) = 0 & z_{Grpr}(6) = 0 \end{cases} \tag{46}$$

where $z(i)$, $i = 1, 2, ..6$, represents the state variable $i$ in state vector $z$. $x_{Ai}$ and $y_{Ai}$ ($i = 1, 2, 3$) are the positions of $A_{irpr}$, while $x_{Grpr}$ and $y_{G \ rpr}$ are the positions of the mobile platform at its mass center, $G$. Substituting the boundary conditions into Equations (41) and (42), then, the driving torque in the inboard state of 3-RPR PPM can be solved with Equation (45).

*3.3. Dynamic Model with Virtual Work Principle*

3.3.1. Jacoby Matrix and Singular Analysis

In general PPMs, the relationship between the velocity $\dot{q}$ of the actuated joint of the PPMs and the velocity $\dot{X}$ of the mass center $G$ of the mobile platform can be expressed in the form of a Jacobi matrix [36], written as:

$$\dot{q} = J\dot{X} \tag{47}$$

where $\dot{X} = \begin{bmatrix} \dot{x}_G, \dot{y}_G, \dot{\varnothing} \end{bmatrix}^T$, while $J$ represents the Jacobian matrix of the PPMs.

Singular configurations result in the end-effectors of parallel manipulators with uncontrollable degrees of freedom, usually in the form of joint-locking or the loss of freedom at the end of the platform [10,37]. Singular configurations in the reachable workspace of a parallel manipulator must be avoided to ensure that the control of the end-effector can be resolved at any point in time. The 3-RRR and 3-PRR PPMs can be determined by whether the velocity Jacobi matrix is an ensemble of the singularity matrices. For 3-RPR PPM, this can be determined by the singularity circle, where the maximum singularity-free workspace is obtained when the base and platform are equilateral triangles [38].

3.3.2. Virtual Work Principle

To verify the developed dynamic models of the three PPMs using the DT-TMM, the dynamics models based on the principle of virtual work are also developed for the dynamic modeling of three PPMs. In this paper, all the components of the three parallel manipulators are in the horizontal plane, within which gravity has no effect on the three PPMs. The dynamic equation of PPMs, based on the virtual work principle, can be written as:

$$\delta q \tau + \delta X_G^T F_P + \sum_{i=1}^n \delta X_i^T F_i = 0 \tag{48}$$

where $\delta X_G$, $\delta q = J \delta X_G$, and $\delta X_i = J_i \delta X_G$ are the virtual displacement of the mobile platform, the driving component, and other components in PPM, respectively. $\tau$ is the torque/force applied on the driving component. $F_P = (-m_P \ddot{x}_G, -m_P \ddot{y}_G, -I_P \ddot{\varnothing})^T$ represents the inertia force of the end effector, in which $m_P$ and $I_P$ are the mass and the moment of inertia of the mobile platform, respectively. $F_i = (-m_i \ddot{x}_i, -m_i \ddot{y}_i, -I_i \ddot{\beta}_i)^T$ is the inertia force of the component $i$, while $m_i$ and $I_i$ are the mass and the moment of inertia of the component $i$. Substituting the virtual displacement into Equation (48), the dynamics equation of PPMs, based on the virtual work principle, can be rewritten as:

$$\tau = -(J^T)^{-1}(F_G + \sum_{i=1}^n J_i^T F_i) \tag{49}$$

Equation (49) is the dynamics model of the parallel manipulator; it can be simplified and the result of rearranging the equation is:

$$\tau = M(X)\ddot{X}_G + C(X)\dot{X}_G \tag{50}$$

where $M(X)$ is the generalized inertia matrix of PPMs and the $C(X)$ is for centripetal force and the Coriolis force matrix.

*3.4. Dynamic Simulation and Verification*

To verify the effectiveness of dynamic modeling using a DT-TMM of three PPMs, a numerical example is given and numerical simulation results with DT-TMM are compared with the results with the dynamic modeling, using the virtual work principle and ADAMS. In the simulations, a circular Cartesian trajectory with a constant orientation and a radius of 0.02 m is used as the trajectory of the moving platform, while the constant orientation is chosen as $\varnothing = 45°$. The trajectory is defined as:

$$x_G = 0.02\cos(\pi t) \ (0 \le t \le 5 \text{ s}) \tag{51}$$

$$y_G = 0.02\sin(\pi t) \ (0 \le t \le 5 \text{ s}) \tag{52}$$

Table 1 shows the related parameters of the 3-RRR, 3-PRR and 3-RPR PPMs. The simulations have been carried out with a time interval $\Delta t = 0.01$ s; the trajectory is set in a safe area, such that the manipulator works without any singularity configurations in 5 s. Figures 6–8 show the simulation results of 3-RRR, 3-PRR, and 3-RPR PPMs, based on DT-TMM, while the simulation results of each configuration are compared with the dynamic modeling results, based on the models from the virtual working principle and ADAMS, respectively.

Table 2 shows the root mean square errors between the different methods of three PPMs, compared with the two methods of DT-TMM and ADAMS simulation, the root mean square error values for all three driving forces are only 0.0002 (3-RRR), 0.0015 (3-PRR) and 0.0003 (3-RPR); the calculated order of magnitude is smaller, which means that the two methods agree very well with each other. All the root mean square error values of the DT-TMM model and the virtual work principle model are 0, which means that the

results are completely consistent. The numerical simulations and comparison verify the effectiveness of the DT-TMM model for the dynamic modeling of the 3-DOF PPMs.

**Table 1.** Parameters of the three PPMs.

| Symbols | Unit | Parameters |
|---|---|---|
| $l_{i1} = 0.1$ | (m) | Length of the first link in the 3-RRR chain |
| $l_{i2} = 0.1$ | (m) | Length of the second link in the 3-RRR chain |
| $l_{i3} = 0.1$ | (m) | Length of the link in the 3-PRR chain |
| $\min l_{i4} = 0.1$ | (m) | Minimal length of the link in the 3-RPR chain |
| $\max l_{i4} = 0.2$ | (m) | Maximal length of the link in the 3-RPR chain |
| $L_P = 0.1$ | (m) | Length of the mobile platform side |
| $L_B = 0.3$ | (m) | Length of the base side |
| $m_s = 0.0027$ | (kg) | Mass of the sliders |
| $m_{l_{i1,2,3}} = 0.0071$ | (kg) | Mass of the links in the 3-RRR and 3-PRR PPMs |
| $m_{l_{i4}} = 0.0142$ | (kg) | Mass of the links in the 3-RPR PPM |
| $m_P = 0.1269$ | (kg) | Mass of the mobile platform |
| $\varnothing = 45°$ | (deg) | Orientation of the platform |

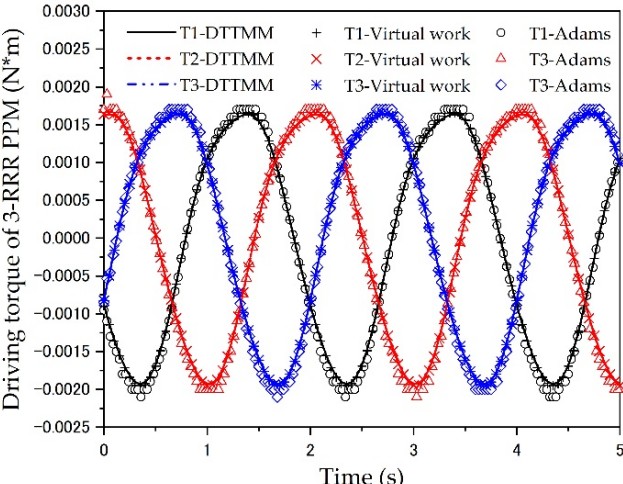

**Figure 6.** Driving torque of the 3-RRR PPM.

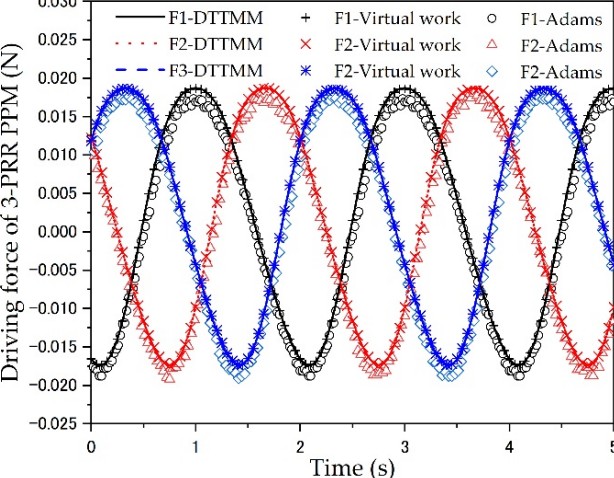

**Figure 7.** Driving force of the 3-PRR PPM.

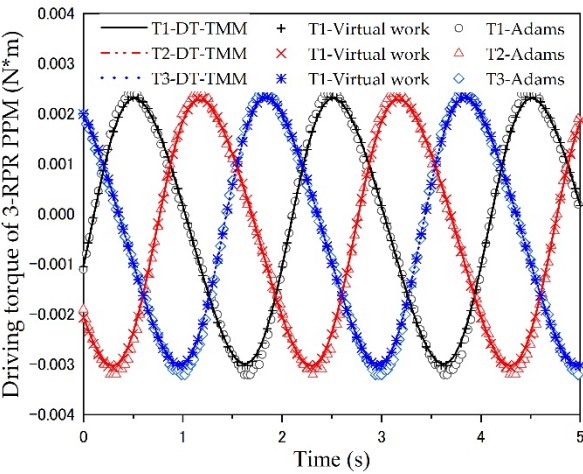

**Figure 8.** Driving torque of the 3-RPR PPM.

**Table 2.** The root mean square error between the different methods of three PPMs.

| Methods | 3-RRR PPM | | | 3-PRR PPM | | | 3-RPR PPM | | |
|---|---|---|---|---|---|---|---|---|---|
| | **T1** | **T2** | **T3** | **F1** | **F2** | **F3** | **T1** | **T2** | **T3** |
| DT-TMM and ADAMS | 0.0002 | 0.0002 | 0.0002 | 0.0015 | 0.0015 | 0.0015 | 0.0003 | 0.0003 | 0.0003 |
| DT-TMM and the virtual work principle | 0 | 0 | 0 | 0 | 0 | 0 | 0 | 0 | 0 |

## 4. Dynamic Performance Indices

The 3-RRR, 3-PRR, and 3-RPR PPMs each have their own advantages and applications. In practical engineering applications, the dynamic performance and dynamic response characteristics of the PPMs should be synthetically evaluated. In this paper, considering the application and working conditions of the parallel manipulators, four dynamic performance indices, i.e., dynamic dexterity index, power requirement, energy transmission efficiency, and joint force/torque margin, are proposed to evaluate and compare the dynamic performances of three PPMs.

### 4.1. Dynamic Dexterity Index

Dynamic dexterity is an important index proposed to evaluate the acceleration and deceleration characteristics of the mechanism [20]. Conventionally, since the inertia factor related to the acceleration plays a crucial role, the generalized inertia ellipsoid is established, based on the generalized inertial ellipsoid principle. The maximum and minimum singular values of the inertia matrix reflect the lengths of the principal axes of the inertia ellipsoid. The ratio of the minimum and maximum singular eigenvalue of the inertia matrix in the dynamic equation, which is reciprocal in terms of the condition numbers $k_D$ of the inertia matrix in the dynamic equation, is adopted as the local dynamic dexterity index for evaluating the dynamic performance and inertia characteristics of the mechanism [39]. $k_D$ is defined as:

$$0 \leq \ k_D = \frac{\sigma_{min}(\boldsymbol{M})}{\sigma_{max}(\boldsymbol{M})} \ \leq 1 \tag{53}$$

where $\sigma_{min}(\boldsymbol{M})$ and $\sigma_{max}(\boldsymbol{M})$ are the minimum and maximum singular values of the inertia matrix $\boldsymbol{M}$. If the lengths of the principal axes are the same, the accelerating performance is isotropic. The difference between the lengths of the major and minor axes represents the anisotropy of the accelerating performance. Thus, the dynamics dexterity index $k_D$ describes the acceleration performance at a given point of the parallel manipulator; when $k_D = 1$, the parallel manipulator has dynamic isotropy.

Considering that $k_D$ varies in the different configurations of the manipulators, global dynamic dexterity can be utilized to describe the mean value of $k_D$ in the workspace. The global dynamic dexterity index is expressed as:

$$0 \leq \bar{\eta}_D = \frac{\int_W k_D dW}{\int_W dW} \leq 1 \tag{54}$$

where $W$ is the task workspace of the parallel manipulator; the closer the index $\bar{\eta}_D$ is to 1, the better the dynamic dexterity and acceleration performance.

### 4.2. Power Requirement

The power requirement of the manipulators, which is directly related to the cost of the system, can be utilized to evaluate the dynamic performance of the parallel manipulator.

The sum of the absolute values of the driving power required for each actuator of the manipulator is used to express the power requirement; this can be written as:

$$P_{sum} = \sum_{i=1}^{n} |P_i| \tag{55}$$

where:

$$P_i = F_i v_i = T_i \omega_i \tag{56}$$

$P_i$ is the power required for each actuator of the PPM, $n = 3$ is the number of driving joints for each PPM, and $F_i$ and $T_i$ are the driving forces and driving torque provided by the actuators, respectively. $v_i$ and $\omega_i$ are the velocity and angular velocity obtained by the prismatic actuators and revolute joint, respectively. When considering the power characteristics, this should be as small as possible to achieve better dynamic performance.

### 4.3. Energy Transmission Efficiency

The smaller the energy loss during the movement of the parallel manipulator, the more efficient its energy transfer will be. To ensure better dynamic performance and increased efficiency of movement, it is essential to analyse and compare the energy transfer efficiency of the PPMs. The effective energy is usually considered as the output energy of the mobile platform, while the ratio of the effective energy to the total energy demand of all the storage components of the parallel manipulator is used as the energy transfer efficiency. The energy transmission efficiency $\eta_E$ is written as:

$$\eta_E = \frac{E_P}{E_{tot}} \tag{57}$$

where the output energy of the moving platform $E_P$ and total energy requirement $E_{tot}$ of the manipulator configuration to accomplish the desired task are written as:

$$E_P = \frac{1}{2}(m_P \dot{x}_G{}^2 + m_P \dot{y}_G{}^2 + I_P \dot{\varnothing}^2) \tag{58}$$

$$E_{tot} = \frac{1}{2}(m_P \dot{x}_G{}^2 + m_P \dot{y}_G{}^2 + I_P \dot{\varnothing}^2) + \frac{1}{2}\sum_{i}^{n}(m_i \dot{x}_i{}^2 + m_i \dot{y}_i{}^2 + I_i \dot{\beta}_i{}^2) \tag{59}$$

where $m_P$ and $I_P$ are the mass and inertia of the moving platform, and $\dot{x}_G$, $\dot{y}_G$, and $\dot{\varnothing}$ are the velocities of the mobile platform, respectively. Similarly, $m_i$ and $I_i$ are the mass and inertia of the other components in each configuration PPM, and $\dot{x}_i$, $\dot{y}_i$, and $\dot{\beta}_i$ are the velocities of the other components in each configuration PPM, respectively. By definition, the higher the energy of the moving platform, the higher the energy transfer efficiency of the PPMs.

*4.4. Joint Force/Torque Margin*

Due to the fact that the driving joints of the three 3-DOF PPMs with different configurations are different, where one is driven by the prismatic joints and the other two are driven by the rotating joints, it is impossible to compare the joint driving force or torque directly. Using a dimensionless performance index, that is, the force/torque margin of the joint force/torque of PPMs, this can be used as the dynamic driving performance, which is one of the most important tools to select the size and appropriate actuators [21,22], written as:

$$p = \sqrt{\frac{1}{n} \sum_{i}^{n} \left( \frac{\tau_i}{\tau_{i\ max}} \right)^2} \tag{60}$$

where $\tau_i$ is the joint force/torques of the actuated joint *i* of the moving platform, and $\tau_{i\ max}$ is the maximum force/torque for the actuated joint *i*. *n* is the number of actuated joints. The force/torque margin of each manipulator becomes larger as the performance index *p* becomes smaller.

## 5. Simulation and Comparison of Dynamic Performances

In this section, the dynamics performance of the three PPMs will be evaluated and compared under both circular and linear trajectory simulations. During all the simulations, the rotation angle of the mobile platform is set to a constant value of $\varnothing = 45°$. To ensure the correct motion of PPMs in the dynamic simulations, singular configurations were excluded. The circular trajectory is the same as that for Equations (51) and (52), while the linear trajectory is a line with a slope of 1, expressed as:

$$x_G = 0.02 + 0.002t \ (0 \leq t \leq 5\ \text{s}) \tag{61}$$

$$y_G = 0.002t \ \ (0 \leq t \leq 5\ \text{s}) \tag{62}$$

In order to obtain better motion performance and make the comparative study more convincing, this section selects the geometric parameters of the three parallel robots, based on the global condition index (GCI). The GCI is the reciprocal of the average number of conditions across the workspace, and it reflects the sensitivity magnitude of the average conditions of the organization in the workspace. The GCI is obtained as:

$$\eta_{GCI} = \frac{\iint_W \frac{1}{k(J)} dW}{\iint_w dW} \tag{63}$$

where the local condition number is $k(J) = \sigma 1 / \sigma 2$. $\sigma 1$ and $\sigma 2$ are the maximum and minimum singular values of the kinematic Jacobi matrix of each PPM.

For a relatively fair performance comparison, we used the three simulated PPMs, except for the fact that the length of the links was selected by GCI, along with the size and weight of the mobile platform and base, and other parameters were considered to be the same [40]. The geometrical parameter with the maximum GCI was chosen as an example of a numerical simulation for three different configurations of PPMs, within the range of the design variables. Table 3 lists the geometric parameters of the maximum GCI for the three PPMs, based on constraints on the mobile platform and base platform.

**Table 3.** The geometric parameters of three PPMs, selected based on the maximum GCI.

| PPM | $L_B$(m) | $L_P$(m) | $l_{i1}$(m) | $l_{i2}$(m) | $\rho_i$(m) | $l_{i3}$(m) | $\min l_{i4}$(m) | $\max l_{i4}$(m) |
|---|---|---|---|---|---|---|---|---|
| 3-RRR | 0.3 | 0.08 | 0.098 | 0.098 | - | - | - | - |
| 3-PRR | 0.3 | 0.08 | | - | 0.03–0.13 | 0.10 | - | - |
| 3-RPR | 0.3 | 0.08 | - | - | - | - | 0.09 | 0.2 |

### 5.1. Comparative Analysis of Dynamic Dexterity

To obtain the global dynamic dexterity, it is necessary to calculate the local dynamic dexterity in terms of the task and in the workspace. The center point $[x_0, y_0]$ of the circular trajectory and the starting point $[x_1, y_1]$ of the linear trajectory are taken as variables. The circular trajectory( $x_G = x_0 + 0.02\cos(\pi t)$ and $y_G = y_0 + 0.02\sin(\pi t)$) and the linear trajectory ($x_G = x_1 + 0.002t$ and $y_G = y_1 + 0.002t$) must be in the workspace, to achieve a normal movement of the parallel manipulator, the singularities should be excluded in the calculation of the workspace [35].

Based on the task workspaces satisfying the circular trajectory and linear trajectory, the distribution of the local dynamic dexterity indices $k_D$ were calculated. Figures 9–11 show the results of $k_D$ under a circular trajectory and Figures 12–14 illustrate the simulation results under a linear trajectory, which demonstrate the distribution of local dynamic dexterity for each PPM under two different trajectories. The color change in the figures corresponds to the change of $k_D$. High local dynamic dexterity indicates a good acceleration. The distribution of the local dynamic dexterity index values is uniform and stable for all three PPMs, and it is evident that there may be a region with the desired dynamics performances when the trajectory center changes from the origin of the coordinates to the edge of the workspace.

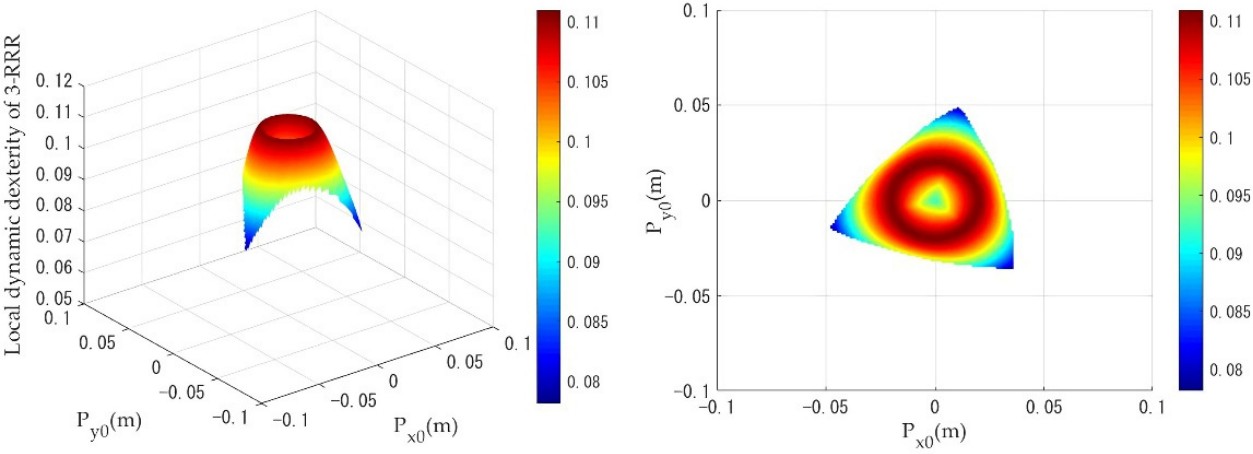

**Figure 9.** Distribution of the local dynamic dexterity of the 3-RRR PPM with a circular trajectory.

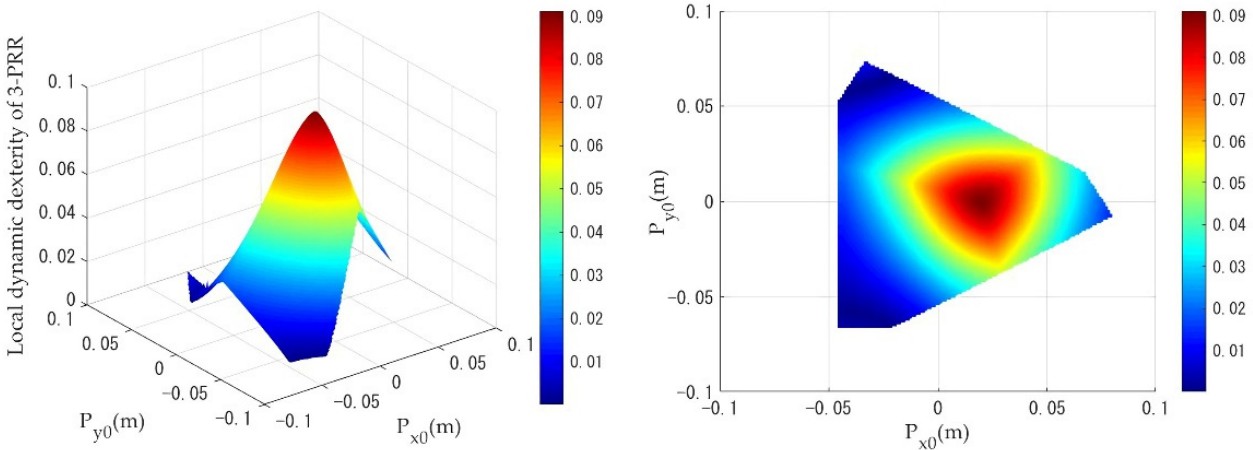

**Figure 10.** Distribution of the local dynamic dexterity of the 3-PRR PPM with a circular trajectory.

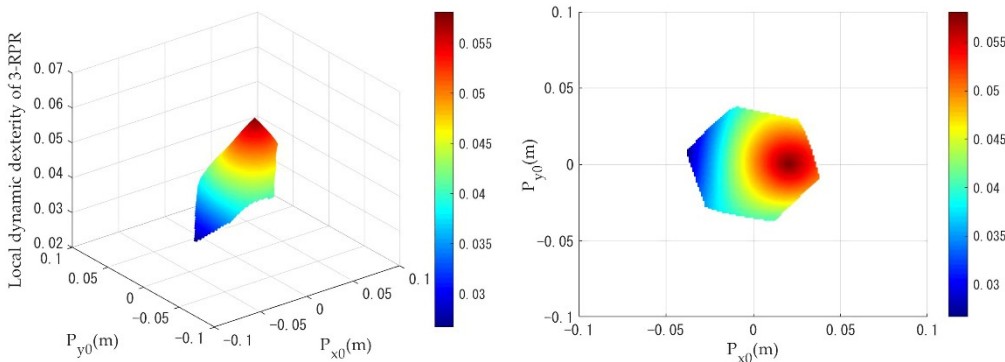

**Figure 11.** Distribution of the local dynamic dexterity of the 3-RPR PPM with a circular trajectory.

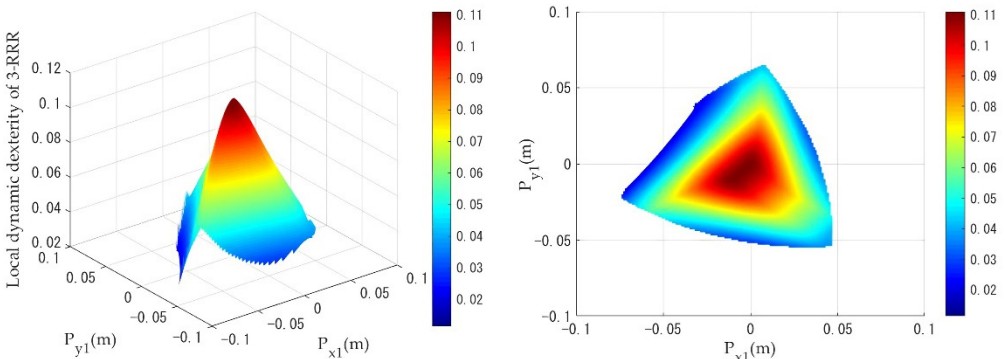

**Figure 12.** Distribution of the local dynamic dexterity of the 3-RRR PPM under a linear trajectory.

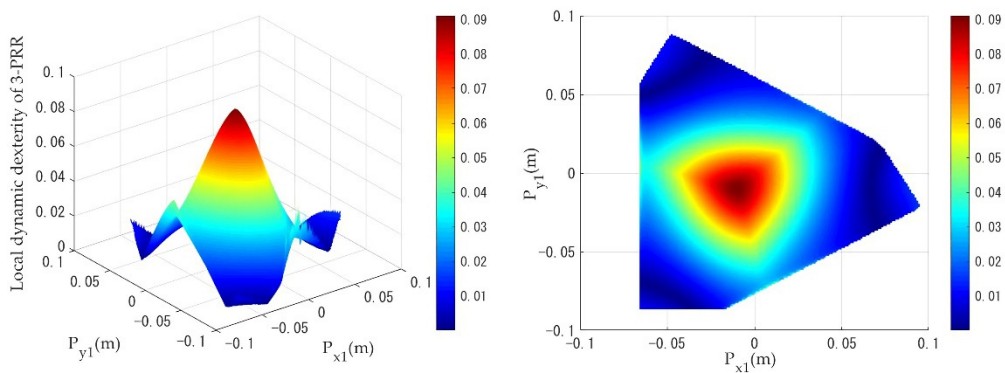

**Figure 13.** Distribution of the local dynamic dexterity of the 3-PRR PPM under a linear trajectory.

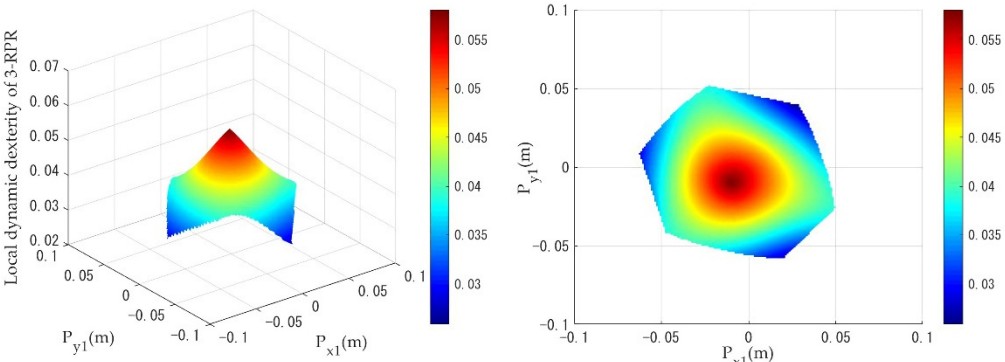

**Figure 14.** Distribution of the local dynamic dexterity of the 3-RPR PPM under a linear trajectory.

When the PPMs are undergoing circular trajectory motion, as shown in Figure 9, the local dynamics dexterity index values $k_D$ of the 3-RRR PPM are evenly distributed from the

center of the workspace toward the edges. As can be seen in Figures 10 and 11, the 3-PRR and 3-RPR PPMs show similar variations in the distribution of local dynamics dexterity index values during motion: the local dynamics dexterity index values reach a maximum for a circular trajectory, with (0.02, 0) as the center, and decline uniformly toward the edges. While the PPMs are undergoing linear trajectory motion, it can be clearly seen from Figures 12–14 that the distribution of the local dynamic dexterity index $k_D$ of the three PPMs has the same trend and all of them reach the maximum in the central region of the task workspace, while the values of the index $k_D$ decrease from high to low, from the central region to the edge of the task workspace.

Table 4 shows the comparison results of global dynamic dexterity for the 3-RRR, 3-PRR and 3-RPR PPMs. The geometrical meaning of $\bar{\eta}_D$ can be interpreted as the mean value of $k_D$; the closer the index $\bar{\eta}_D$ is to 1, the better the dynamic dexterity and acceleration performance. When $k_D > \bar{\eta}_D$, this means that there is better dynamic dexterity in these workspaces and, from this area, it is possible to inform the selection of trajectory movements with good dynamic dexterity for PPMs. Table 4 also gives the percentage of better dynamic dexterity areas on the respective task workspace for the three different PPMs. As shown in Table 4, when the mobile platform performs a circular trajectory motion, the 3-RRR PPM has the highest value of the index $\bar{\eta}_D$ among the three PPMs, which means that it demonstrates the best dynamic dexterity performance under the circular trajectory simulation. In addition, the 3-RRR PPM has 56% better dynamic dexterity in the area of the given task workspace. The value of the index $\bar{\eta}_D$ for the 3-PRR PPM is slightly larger than the 3-RPR PPM, while the better dynamic dexterity area percentage for the 3-PRR PPM is 72%, compared to only 52% for the 3-RPR PPM, representing the larger area available in the workspace of the 3-PRR design. While the mobile platform makes a linear trajectory motion, the 3-PRR PPM has the highest value of the index $\bar{\eta}_D$ among the three PPMs; the better dynamic dexterity area percentage is also the largest, while the value for the 3-RRR PPM is slightly smaller. Thus, the 3-PRR PPM shows the best dynamic dexterity performance under the linear trajectory simulation. Additionally, it can be seen that the value of the index $\bar{\eta}_D$ for 3-RPR PPM is the smallest under both trajectories, and the better dynamic dexterity area percentage is also the smallest. Therefore, the 3-RPR PPM has the worst dynamic dexterity performance.

**Table 4.** Comparison of the global dynamic dexterity of three PPMs.

| PPM | Circular Trajectory | | Linear Trajectory | |
|---|---|---|---|---|
| | $\bar{\eta}_D$ | $Area_{k_D > \bar{\eta}_D}$ (%) | $\bar{\eta}_D$ | $Area_{k_D > \bar{\eta}_D}$ (%) |
| 3-RRR | 0.1015 | 56.31% | 0.0602 | 45.53 |
| 3-PRR | 0.0592 | 73.37% | 0.0631 | 69.95 |
| 3-RPR | 0.0438 | 52.13% | 0.0452 | 46.25 |

*5.2. Comparative Analysis of Power Requirement*

In this section, the power requirements of the three PPMs are calculated using Equation (55). The comparison results of the power requirements among the three PPMs under two different trajectories are shown in Figure 15.

As seen from Figure 15a, under the circular trajectory motion, the 3-PRR manipulator has the largest range of the sum of the absolute values of the required driving power, while the 3-RPR manipulator ranks in second place. The 3-RRR parallel manipulator has the smallest range of required drive power among the three planar 3-DOF parallel mechanisms. That means that the 3-RRR PPM requires substantially less power when undergoing circular trajectory motion. When the simulation is performed for the linear trajectory, it can be seen in Figure 15b that the 3-PRR PPM has the smallest range of driving power requirement among the three PPMs, while the 3-RPR PPM has the highest power requirement.

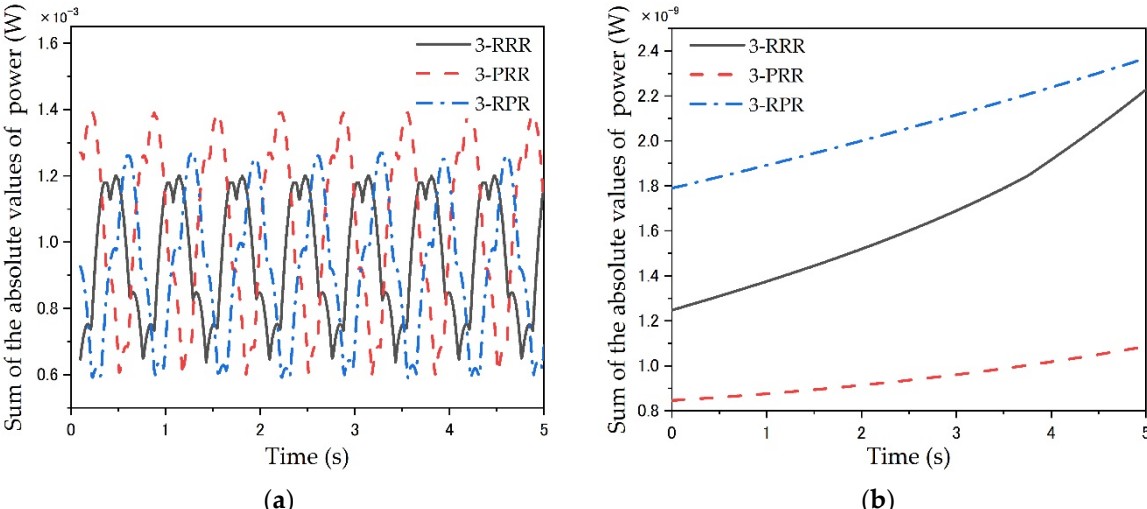

**Figure 15.** The sum of the absolute values of the power requirements of three PPMs: (**a**) for the circular trajectory; (**b**) for the linear trajectory.

### 5.3. Comparative Analysis of Energy Transmission Efficiency

The energy transfer efficiency of the three PPMs is calculated using Equation (57) under two different trajectory motions. The higher the energy transfer efficiency, the lower the energy loss during the transfer process, which means that the dynamic transfer performance of the parallel manipulator is better.

Figure 16 shows the comparison results of the energy transfer efficiency of the three PPMs under the two trajectories. As shown in Figure 16a,b, the comparison results of energy transfer efficiency under both two trajectories are the same. The 3-RPR PPM has the highest energy transfer efficiency, with mean values of 0.9200 and 0.9197 under circular and linear trajectories, respectively. The 3-RRR PPM has slightly lower energy transfer efficiency values than the 3-RPR PPM, with mean values of 0.8805 and 0.8781. However, the 3-PRR PPM has the smallest energy transfer efficiency values, with mean values of 0.7745 and 0.7962. Therefore, in the dynamic simulations when undergoing two different trajectories, the 3-RPR PPM has the best energy transfer performance, while 3-PRR PPM has the worst.

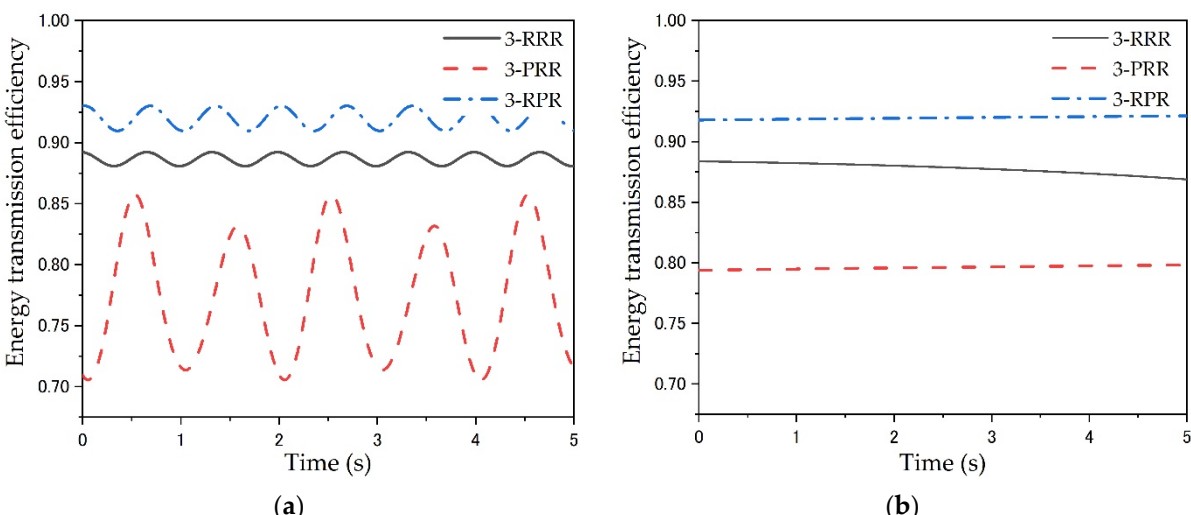

**Figure 16.** Comparison of the energy transmission efficiency of three PPMs: (**a**) under a circular trajectory; (**b**) under a linear trajectory.

### 5.4. Comparative Analysis of the Joint Force/Torque Margin

Similarly, the driven forces and torques calculated in the circular trajectory and linear trajectory simulations are used to obtain the dynamic performance index $p$, to evaluate and compare the joint force/moment margins among the three PPMs. The driving performance of one manipulator becomes better as the dynamic performance index $p$ becomes smaller.

The comparison results of the dynamic performance index $p$ for the three PPMs under a circular trajectory is shown in Figure 17a. It can be found that the dynamic performance index value $p$ of 3-RPR PPM is the smallest compared to other configurations. This means that it has the best driving performance, while both the 3-RRR and 3-PRR PPMs have worse joint force/torque margins than the 3-RPR PPM. The largest value of $p$ for 3-RRR PPM also indicates the worst drive performance. Figure 17b shows the comparison results of the values of the exponent $p$ for the three PPMs under the linear trajectory. The 3-PRR has the smallest $p$ among the three PPMs and has the best driving performance under the linear trajectory. Comparing the 3-RRR and 3-RPR PPM, the mean value of the $p$-value for 3-RRR PPM is the largest at 0.7175, while the mean value of the 3-RPR PPM is 0.7150. Thus, the driving performance of the 3-RRR PPM under the linear trajectory is also the worst.

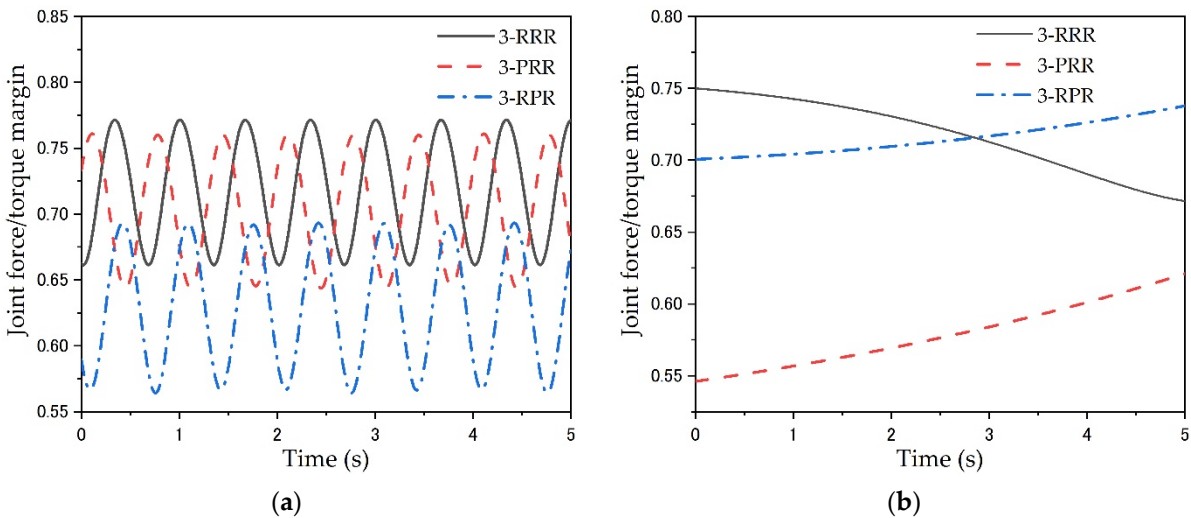

**Figure 17.** Comparison of the joint force/torque margin of the three PPMs: (**a**) under a circular trajectory; (**b**) under a linear trajectory.

## 6. Conclusions

In this paper, the dynamic performance of three manipulator configurations, namely, the planar 3-DOF 3-RRR, 3-PRR, and 3-RPR parallel manipulators, have been investigated and compared. The dynamic modeling formulations based on DT-TMM have been established and verified via dynamic modeling, using the virtual work principle and ADAMS 2016 software. The performance evaluation indices, including dynamics dexterity, the power requirement, the energy transmission efficiency, and the joint force/torque margin, have been proposed to compare their dynamic performances under a general circular trajectory and a linear trajectory. The simulation results show that compared with the other two configurations of PPMs, the 3-RRR PPM has advantages under circular trajectory: having the best dynamic dexterity performance, the smallest power demand range, and the second-highest energy transfer efficiency. It is, therefore, more suitable for the field of medical robotics and micro-manipulation. The disadvantage of the 3-RRR is that it has the worst joint force/torque margin under both trajectory motions. In contrast, the 3-PRR PPM has great advantages when undergoing a linear trajectory: it offers the best dynamic dexterity, has the smallest power requirement range, and provides the best drive performance. It has more excellent application prospects in the field of mechanized automatic production, packaging, and transportation. However, the disadvantage of 3-PRR is the lowest energy transfer efficiency. Among the three PPMs, the 3-RPR PPM has the highest energy transfer

efficiency and has better dynamic performance in a circular trajectory than a linear motion; therefore, it can be used for an industrial robot.

**Author Contributions:** Conceptualization, X.Z. and G.S.; data curation, F.C.; formal analysis, G.S. and F.C.; investigation, X.Z. and G.S.; methodology, X.Z.; project administration, G.S.; software, F.C.; supervision, G.S. and X.Z.; validation, G.S. and F.C.; visualization, F.C.; writing—original draft, F.C.; writing—review and editing, G.S. and X.Z. All authors have read and agreed to the published version of the manuscript.

**Funding:** This research received no external funding.

**Institutional Review Board Statement:** Not applicable.

**Informed Consent Statement:** Not applicable.

**Data Availability Statement:** The data generated during the current study are available from the corresponding author on reasonable request.

**Conflicts of Interest:** The authors declare no conflict of interest.

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
