# Peer review of "Comparison of the Dynamic Performance of Planar 3-DOF Parallel Manipulators"

_machines, doi:10.3390/machines10040233_

Round 1
Reviewer 1 Report
This paper addresses a quite interesting topic, such as the comparison of performance in three different types of parallel planar manipulators (PPM). The criteria used for comparison are: dynamic dexterity, power requirements, energy transmission efficiency and force/torque margin in kinematic pairs. However, some important concerns should be addressed before publication.
- About half of the paper is devoted to obtaining the dynamic model of the PPR’s using the discrete transfer matrix method, which is not directly related to the title of this paper. Indeed, said dynamic model can be obtained in an exact and computationally efficient way by other standard procedures. Taking into account the simplicity of the robots, the computation time required for a dynamic model does not seem to be a sufficient reason for such an extensive development of an alternative method for computation of the dynamic model.
- Another important aspect that is not completely defined is what type of dynamic analysis is performed in Section 5. As indicated in line 404, only the position of the center of the mobile platform is established as a function of time. The questions that arise at this point are:
- What happens with the orientation of the mobile platform?
- Is it, as it seems, performing an Inverse Dynamic Analysis?. If the comparison of the dynamic performances were being carried out using an Inverse Dynamics, this could be done very easily using the usual dynamic analysis methods.
In short, it is suggested to the authors:
- Strongly argue the need to use this procedure to perform the dynamic simulation of the PPM
- If the case, significantly simplify the obtaining part of the dynamic modelling (Section 3)
- It is well known that one of the main problems faced by parallel robots is the presence of singularities, specifically Type II singularities within their workspace. However, this paper does not refer to this problem. It is only mentioned that the trajectories used are free of singularities and in line 391 a Jacobian matrix is mentioned, without specifying if it refers to the Forward Jacobian.
- The third issue that needs to be addressed is that the comparison of the PPM performances is done using a single trajectory. The doubt arises if the conclusions are valid for other trajectories different from the one considered. The authors must justify how their conclusions are general based on a single trajectory, or carry out tests with other trajectories clearly different from the one indicated in the paper.
Minor remarks:
Taking into account that major changes to the text have been suggested, it is considered more appropriate to leave minor changes for a next round of revision.
Reviewer 2 Report
This paper investigates the dynamic performance comparison between three 3-DOF parallel manipulators, and simulations results are provided. The work is interesting, and some suggestions are given as follows:
- What’s the main advantage of the DT-TMM compared to the virtual work principle method?
- It is unfair to compare the dynamic performance with different workspaces.
- The symbol of vector should be bold and italic.
- Based on the comparison results, could the author give some appropriate application scenarios of these three parallel manipulators?
- The following papers about parallel manipulators could be cited:
1) Minimum-time trajectory planning and control of a pick-and place five-bar parallel robot. IEEE/ASME Transactions on Mechatronics, 2015, 20(2): 740-749.
2) An experimental study on the dynamics calibration of a 3-DOF parallel tool head. IEEE/ASME Transactions on Mechatronics, 2019, 24(6): 2931-2941.
Round 2
Reviewer 1 Report
The changes introduced by the authors adequately respond to the issues raised in the previous review; consequently, the publication of the article proceeds in its present form
Reviewer 2 Report
The paper can be accepted.